



**Variations of carbon flux at different time scales in a semi-fixed sandy**
**land ecosystem in Horqin Sandy Land, China**
Yayi Niu[a,c,d], Yuqiang Li[a,b,c,d]*, Wei Liu[a,c], Xuyang Wang[a,c,d], Yun Chen[a,c,d]
[a] Northwest Institute of Eco-Environment and Resources, Chinese Academy of
Sciences, Lanzhou 730000, China
[b] Key Laboratory of Strategic Mineral Resources of the Upper Yellow River, Ministry
of Natural Resources, Lanzhou 730000, China
[c] University of Chinese Academy of Sciences, Beijing 100049, China
[d] Naiman Desertification Research Station, Northwest Institute of Eco-Environment
and Resources, Chinese Academy of Sciences, Tongliao 028300, China
* *Correspondence to*: Yuqiang Li (liyq@lzb.ac.cn)
320 Donggang West Road, Lanzhou, 730000, China
Phone/Fax: 86-931-496-7219





**Abstract**

Sandy land is an important part of terrestrial ecosystem, which has a substantial
impact on maintaining global ecological health and security. However, there is still a
scarcity representative studies of climate change's effect on the carbon fluxes (*NEE*:
net ecosystem $CO_2$ exchange; $R_{eco}$: ecosystem respiration; *GPP*: gross primary
productivity). Eddy covariance technique was used to determine carbon fluxes and
climatic conditions in this ecosystem from 2017 to 2021. At an annual scale, the semi-
fixed sandy land was found to be a net carbon release, the value of annual average *NEE*
was $6.81 \pm 36.35$ g C $m^{-2}$ $yr^{-1}$. It functioned as a carbon source in dry years (2017 and
2020), but was a carbon sink in wet years (2019 and 2021) and a normal year (2018).
At seasonal scale, according to the Random Forest, deep soil water content ($SWC_{80}$)
and photosynthetic photon flux density had a great impact on *NEE* and *GPP*, whereas
shallow and deep soil water ($SWC_{10}$ and $SWC_{80}$) dominated $R_{eco}$. At a monthly scale,
the multiple stepwise regression showed that soil temperature, precipitation (*PPT*) and
*SWC* were the dominant environmental factors. At an annual scale, correlation analysis
showed that total annual *PPT* was negatively correlated with *NEE*. Our results illustrate
the importance of climate variations for the *NEE*, $R_{eco}$, and *GPP* at different time scales
in arid and semi-arid areas. They also highlight the importance of water availability (the
pattern and intensity of *PPT* and *SWC* at different depths) on regional and global carbon
cycles.
**Keywords:** Precipitation; Net ecosystem $CO_2$ exchange; Carbon flux; Climate change;
Semi-fixed sandy land
**1. Introduction**

Human activities have led to unprecedented and devastating global climate change,

including altered precipitation patterns, increased temperature and $CO_2$ concentration
(IPCC, 2007; Yu et al., 2013). $CO_2$ plays a major role in Earth's mass and energy
budgets, so quantifying ecosystem carbon cycles and carbon budgets is essential for
planning a sustainable future (McGuire et al., 2009; Ma et al., 2020). The balance
between photosynthesis (gross primary productivity (*GPP*)) and respiration (ecosystem



respiration ($R_{eco}$)) determines the net ecosystem $CO_2$ exchange (*NEE*) in terrestrial
ecosystems (Schmitt et al., 2010; Lasslop et al., 2010; Zhang et al., 2019). As a result,
comprehending the dynamics processes and underlying mechanisms of *NEE* is a crucial
issue in global change research (Yu et al., 2013). In particular, the division of *NEE* into
*GPP* and $R_{eco}$, which depict the fundamental mechanisms, is conducive to supply a
process-level, mechanistic comprehension of the regional carbon balance (Reichstein
et al., 2005; Jassal et al., 2007; Lasslop et al., 2010; Cao et al., 2021).
About 30% of the earth's surface is covered by arid and semiarid areas (Kefi et al.,
2008; Poulter et al., 2014). Ecosystems in these areas are at risk of soil erosion and
degradation, which is resulting from a combination of climate change, such as
increasing intensity and frequency of extreme climate events (eg., drought and extreme
precipitation) (Knapp et al., 2015), and unreasonable human activities, including
overgrazing, firewood harvesting, and excessive deforestation (Domingo et al., 2011;
Wang et al., 2021). Despite the fact that many research on carbon fluxes through
ecosystems in semi-arid areas have been conducted to date (Du and Liu, 2013; Hao et
al., 2017; Niu et al., 2020; Zhang et al., 2020), we still don't fully understand how these
ecosystems function as $CO_2$ sources or sinks (Huxman et al., 2004; Ma et al., 2007;
Zhou et al., 2020; Niu et al., 2021). Therefore, further investigation is required to study
the carbon budget and its controlling mechanisms for a wider range of ecosystems,
particularly in drylands, to more accurately evaluate global carbon budgets.
The original landscape of China's Horqin Sandy Land was a sparse shrubland and
grassland with hydrothermal conditions that are adequate to support shrubs, an
abundant plant community with a diverse composition, a stable ecosystem structure,
and high productivity. However, due to the disturbance caused by extensive human
activities and the cultivation of a large area of land, nearly 80% of the region has
experienced aeolian desertification (Liu et al., 1996; Zhao et al., 2007; Li et al., 2019;
Niu et al., 2020). The region's eco-environment is extremely fragile and vulnerable to
damage (Meng et al., 2008; Zhu et al., 2020). Grassland, cropland, and ecosystems with
semi-fixed sands are the main land uses in this study area (Duan et al., 2019; Zhu et al.,
2020). Previous researches in this area have shown that the recovering sandy grassland



ecosystem was a carbon release, whereas the sandy maize cropland ecosystem was a
carbon sink on an annual scale, and that the amount of carbon sequestration in both
ecosystems increased with increasing precipitation (Niu et al., 2020, 2021). To the best
of our knowledge, the carbon fluxes of this semi-fixed sandy land ecosystem in this
region have received little attention (e.g., Hu et al., 2015), so more studies are needed
to understand the characteristics of carbon flux at the ecosystem scale, especially for
semi-fixed sandy land protected by grazing exclusion using fences to allow natural
recovery. Long-term continuous monitoring of the carbon flux characteristics and the
influencing factors in the semi-fixed sandy land will complement and improve our
understanding of the present carbon budgets of the world's dryland ecosystems.
Studies have found that  numerous meteorological factors can play a vital importance
in regulating carbon flux (*NEE, GPP* and $R_{eco}$) (Hu et al., 2010; Papale et al., 2015;
Tang et al., 2018; Zhang et al., 2018a; Watham et al., 2021). For example, previous
studies have shown that photosynthetic photon flux density (*PPFD*), air and soil
temperatures, precipitation (*PPT*), soil water content (*SWC*), and vapor pressure deficit
(*VPD*) most strongly controlled the dynamics of ecosystem carbon fluxes (Jia et al.,
2014, 2016; Liu et al., 2019). However, the availability of water is generally the main
constraining factor that affects the carbon flux characteristics of the system in water-
limited ecosystems (Fu et al., 2006; Zhang et al., 2019; Zhou et al., 2020; Niu et al.,
2020, 2021), so water-related parameters like *PPT*, *SWC*, and *VPD* have a significant
impact on the variations in carbon fluxes at different temporal scales (Noormets et al.,
2010; Gao et al., 2012; Jia et al., 2014; Niu et al., 2020). Both *GPP* and $R_{eco}$ may be
limited by low water availability (Yuan et al., 2010; Zhou et al., 2013; Zhang et al.,
2018). The depression of *GPP* results from limitations on plant physiological processes
and alterations of plant phenology (Meir and Woodward, 2010; Zhou et al., 2013),
whereas the depression of $R_{eco}$ results from decreased root respiration (Linn and Doran,
1984; Bouma et al., 1997; Lee et al., 2003), decreased soil microorganism activity
(Skopp et al., 1990; Drenovsky et al., 2004), and decreased decomposition of organic
matter (Liu et al., 2009; Moyano et al., 2012; Cuevas et al., 2013; Wang et al., 2014).
Therefore, quantifying how precipitation affects soil water regimes, and how these





changes influence *NEE*, *GPP* and $R_{eco}$, is critical to evaluate the vulnerability of sandy
land ecosystems to climate change, which will be important information to support
development of strategies to preserve or restore these sandy lands (Zhang et al., 2019;
Niu et al., 2020).
In this paper, we concentrated on carbon fluxes (*NEE*, $R_{eco}$, and *GPP*) from 2017 to
2021 in a semi-fixed sandy land ecosystem in Inner Mongolia, China. Our main goals
were to (1) quantify the inter-annual, seasonal, and monthly changes of carbon fluxes
(*NEE*, $R_{eco}$, and *GPP*); (2) identify the environmental variables that control these
variations at the different time scales; and (3) given that *PPT* and *SWC* are dominant
factors that influence carbon fluxes in sandy maize cropland and sandy grassland
ecosystems in this region (Niu et al., 2020, 2021), identify the impact of *PPT* and *SWC*
on carbon fluxes in the semi-fixed sandy land ecosystem. Our research hypothesis was
that changes in climate factors at different temporal scales, and particularly changes in
*PPT* and *SWC*, would affect *NEE*, *GPP,* and $R_{eco}$ through both direct and indirect
mechanisms.
**2. Materials and methods**
**2.1 Site description**
The research was conducted at the Horqin Sandy Land in Naiman Banner, Tongliao
City, Inner Mongolia, China (42° 55′N, 120° 42′ E, 377 m a.s.l.) (Fig. 1)). The study
site became severely desertified due to over reclamation and overgrazing. A series of
grazing exclosures were erected in 2005 to restore the function of degraded ecosystem.
The climate features of the site are temperate continental semiarid monsoon climate,
with average annual temperature was 6.8 °C, varying from −9.6 °C in January and
24.6 °C in July. The mean annual precipitation was 360 mm, a large portion of annual
precipitation (70 %) occurs from May to September (Niu et al., 2020, 2021). The soil
in this site are chestnut soil and Aeolian sandy soil (Zhao et al., 2007; Niu et al., 2020).
Soil texture of topsoil (0-20 cm) is comprised by 92% coarse sand, 2% fine sand, and
6% clay, respectively. Other soil properties such as pH, soil organic carbon, bulk
density, total nitrogen contents, and field capacity were 7.42, 2.47 g kg$^{-1}$, 1.66 g cm$^{-3}$,
0.16 g kg$^{-1}$, and 24.5%, respectively. Vegetation basal cover ranges from 30% to 60%,



dominated by *Caragana microphylla* in the shrub layer, and the herbaceous species

included *Setaria viridis*, *Pennisetum centrasiaticum*, *Chloris virgata*, and *Artemisia*

*scoparia*.

**2.2 Carbon fluxes and micrometeorological measurements**

The eddy covariance (EC) method was utilized to determine $CO_2$ flux at half-hourly

intervals from 2017 to 2021. *PPFD*, air temperature ($T_a$), precipitation (*PPT*), soil heat

flux at two depths (5 and 10 cm), soil water content (*SWC*) and the soil temperature ($T_s$)

at four depths (10, 30, 50, and 80 cm) were also measured. A complete description of

the equipment used and protocols for eddy covariance data processing (including raw

10 Hz data, 30-min data quality and gap filling methods) are described in Niu et al.

(2020; 2021). The degree of energy closure was used to assess the data quality of an

EC system (Wilson et al. 2002). The energy closures ranged from 0.48 to 0.67

throughout our study (Fig. S1), indicating that the data observed at our study site met

the observation requirements (Wilson et al. 2002; Niu et al., 2021).

**2.3 Random Forest and statistical analyses**

The Random Forest analysis was used to identify the main influencing factors of

seasonal *NEE*, $R_{eco}$ and *GPP* among the meteorological factors (*PPT*, *PPFD*, $T_a$, *VPD*,

the $T_s$ and *SWC* at four depths) (Pham and Brabyn, 2017). Because the root system of

the natural *Caragana microphylla* shrubs is mainly found above a depth of 80 cm (A et

al., 2003), we measured the *SWC* and $T_s$ at depths of 10, 30, 50, and 80 cm. More

detailed procedures for Random Forest can found in Zhou et al. (2020) and Niu et al.

(2021).

We also used multiple stepwise regression analysis to identify the key environment

factors (*PPT*, *PPFD*, $T_a$, *VPD*, and the $T_s$ and *SWC* at depths of 10, 30, 50 and 80 cm)

linked to monthly scale *NEE*, $R_{eco}$ and *GPP*. The higher *F*-values represent a better fit.

For the data at an inter-annual scale, the relationship between the climatic factors and

the carbon fluxes was determined using correlation analysis (Pearson's *r*). We used

SPSS (22.0 version Inc. Chicago, IL) software to perform all descriptive statistics and

statistical analyses (including multiple stepwise regression analysis, single-factor

analysis of variance (ANOVA), and correlation analysis). We used least-significant-





different (LSD) test to determine pairs of values that differed significantly. We used
version 8.0 of the Origin software (OriginLab Corporation, Northampton, MA, USA)
for graphing our results.
**3. Results**
**3.1 Meteorological conditions**
Environmental factors showed apparent seasonal variations (Fig. 2). Temperature ($T_a$
and $T_s$) and $PPFD$ both followed unimodal type distribution. During the observation
period, the mean annual $PPFD$ was 22.14 mol m$^{-2}$ d$^{-1}$ (Table 1), with daily values
ranging from 1.89 mol m$^{-2}$ d$^{-1}$ on 13 February 2020 to 48.44 mol m$^{-2}$ d$^{-1}$ on 15 June
2020. The $PPFD$ was significantly lower in 2020 (1544 mol m$^{-2}$ d$^{-1}$) than in the other
years, and there was no difference among the other years. The mean annual $T_a$ was 8.2 °C
(Table 1), with daily values ranging from -23.61 °C on 7 January 2021 to 32.16 °C on
3 August 2021. The mean annual $T_{s10}$, $T_{s30}$, $T_{s50}$, and $T_{s80}$ were 9.94, 10.26, 10.63, and
10.97 °C, respectively, with daily values of $T_{s10}$ ranging from -18.74 °C on 27 January
2018 to 36.04 °C on 26 July 2020, daily values of $T_{s30}$ ranging from -13.92 °C on 27
January 2018 to 36.67 °C on 25 July 2020, daily values of $T_{s50}$ ranging from -9.89 °C
on 28 January 2018 to 27.61 °C on 16 July 2017, and daily values of $T_{s80}$ ranging from
-7.97 °C on 25 January 2017 to 25.72 °C on 6 August 2018. The mean annual $T_a$ did
not differ appreciably across years, but the mean annual $T_s$ did not differ significantly
among the years at any depth ($P > 0.05$).
The annual cumulative $PPT$ varied greatly during the observation period from 2017
to 2021, and differed significantly between many pairs of years (Fig. 2f and Table 1):
it averaged 313 mm in 2017, 351 mm in 2018, 382 mm in 2019, 312 mm in 2020, and
430 mm in 2021. As a result, the $PPT$ in 2017 and 2020 were less than in a typical year
(long-term average of 360 mm for 1960-2014; Niu et al., 2020), whereas 2018 was near
to a normal year. $SWC$ followed the same general trend as $PPT$ (Fig. 2e).
$VPD$ is also related to $PPT$, but it has the opposite pattern as $SWC$. $VPD$ showed a
unimodal type (Fig. 2a). However, the mean value did not differ significantly among
the years, except for a significantly higher value in the dry year 2017 (Table 1).



## 3.2 Variations in carbon fluxes

*NEE*, $R_{eco}$, and *GPP* showed obvious seasonal changes throughout the growing season (from May to September), whereas *NEE* was generally stable across years outside the growing season (Fig. 3). Daily *NEE*, $R_{eco}$, and *GPP* showed resemble seasonal dynamics during the whole study period, and only a few days during the growing season showed net carbon emission; the rest showed carbon absorption. However, the size of *NEE*, $R_{eco}$ and *GPP* varied throughout research years.

At a monthly scale (Fig. 4a-e), $R_{eco}$ and *GPP* generally showed unimodal trends and peaked in July. An exception was the June peak in 2020, a year when *PPFD* was significantly lower than in all other years; Table 1), with lower $R_{eco}$ and *GPP* all year compared with the other years (Fig. 4d). Due to the influence of *PPT* and *SWC* on these fluxes, *NEE* in the wet years showed carbon absorption throughout the growing season (from May to August in 2019 and 2021; Fig. 4c, e; Table 1), whereas in the dry and not significantly different from the long-term average close to a normal year, *NEE* showed carbon absorption in about 3 months and carbon emission in the other months (2017, 2018 and 2020; Fig. 4a, b, d; Table 1).

At an annual scale (Fig. 4f), this study shows that the mean *NEE*, $R_{eco}$, and *GPP* were 6.81 ± 36.35, 664.78 ± 31.49, and 658.79 ± 46.11 g C m$^{-2}$ yr$^{-1}$, respectively. In the wet years (2019 and 2021), the semi-fixed sandy land showed carbon sequestration, the cumulative annual *NEE* were -14.14 and -126.14 g C m$^{-2}$ yr$^{-1}$, respectively. In contrast, in the dry years (2017 and 2020) and the normal year (2018), the system showed carbon emissions, the cumulative annual *NEE* were 48.50, 51.17, and 74.66 g C m$^{-2}$ yr$^{-1}$, respectively.

## 3.3 Relationships between meteorological factors and *NEE*, $R_{eco}$ and *GPP*

The environment factors and *NEE* were largely stable during dormant season in all years, so in the rest of this paper, we will concentrate on the relationships between the carbon fluxes and the meteorological factors during the growing season. Figure 5 illustrates the variable importance values from the Random Forest analysis, which represent the contributions of the variables to *NEE*, $R_{eco}$, and *GPP*. The goodness of fit measure of the random forest analysis is shown in Fig. S2. For *NEE*, the most critical



variable was *PPFD*, with an importance of 68.6%, followed by the factors associated
with moisture (44.4% for $SWC_{80}$, 39.3% for *PPT*, 36.7% for $SWC_{30}$, 34.8% for $SWC_{50}$,
*VPD* for 34.0%, and 27.1% for $SWC_{10}$), which were all significant at $P < 0.01$. For $R_{eco}$,
the soil shallow *SWC* ($SWC_{10}$) and deep *SWC* ($SWC_{80}$) were the most important
variables, with importance values of 73.4% and 65.1%, respectively, followed by
temperature (61.2% for $T_a$ and 46.9% for $T_{s50}$), $SWC_{30}$ (41.0%), and *PPFD* (27.63%),
which were all significant at $P < 0.01$ (except for $SWC_{30}$, which was significant at $P <$
0.05). For *GPP*, *PPFD* was the most important factor, with an importance of 67.48%,
followed by the factors associated with moisture (55.3% for $SWC_{80}$, 50.7% for $SWC_{50}$,
and 44.6% for $SWC_{30}$, and 48.3% for $SWC_{10}$). These were followed by two temperature
variables ($T_{s50}$ and $T_a$ and 44.0% and 36.7%), which were significant at $P < 0.01$. In
general, *PPFD*, deep soil moisture ($SWC_{80}$) and shallow soil moisture ($SWC_{30}$ and
$SWC_{10}$) were the main environmental factors that affected all three carbon fluxes at a
seasonal scale, and they showed a strong and negative relationship with *NEE* and a
significant positive relationship with $R_{eco}$ and *GPP* (Fig. 6); that is, the ecosystem's
carbon sequestration potential rose as *PPFD* and *SWC* increased.

We used the multiple regression analysis to reveal the relationships between the *NEE*,

$R_{eco}$, and *GPP* and environmental parameters to determine the main influencing factors
that resulted in the monthly variation in carbon fluxes. The results are summarized in
Table 2. We found that 65% of the *NEE* variation could be explained by a combination
of $T_{s50}$, $SWC_{30}$, *VPD*, and *PPFD* ($F$=28.75, $R^2$=0.65, $P < 0.001$). We found that 83% of
the variations of $R_{eco}$ could be explained by a combination of $T_{s10}$, *PPT*, and $SWC_{10}$
($F$=91.96, $R^2$=0.83, $P < 0.001$). Similarly, 85% of the variations of *GPP* could be
described by a combination of $T_{s10}$, *PPT*, $SWC_{80}$, and $SWC_{10}$ ($F$=82.62, $R^2$=0.85, $P <$
0.001). Generally, $T_{s10}$, *PPT*, and *SWC* explained a large amount of the carbon flux
variations. The correlations between the key variables and the carbon fluxes were
highest for $T_{s10}$, *PPT*, and *SWC* in each model (Figs. 7-9).

At the yearly scale, the mean *PPFD*, $T_a$, $T_s$ at all depths, and *VPD* were relatively

stable across the study period, and the relationships between these environment factors
and the *NEE*, $R_{eco}$, and *GPP* were not significant. The main environmental variation



during the study period was the availability of water, such as *PPT* and the *SWC* at all
depths (Fig. 2, Table 1). We found that *NEE* strong and negative related with *PPT*, but
not related with the other environmental factors (Table 3); $R_{eco}$ didn't have a significant
relationship with environmental conditions, and *GPP* was significantly positively
related with *PPFD* and $SWC_{80}$. That is, the ecosystem's carbon sequestration capacity
rose when *PPT*, *PPFD*, and $SWC_{80}$ increased.
**4. Discussion**
**4.1 Comparison with other dryland ecosystems**
The ecosystem of *NEE* changes largely from carbon sequestration to carbon
emissions, and these changes generally rely on water availability in dryland ecosystems
(Mielnick et al., 2005; Liu et al., 2012). Our study showed that the system in semi-fixed
sandy land was a net carbon emission in dry years, and a weak carbon absorption in
relatively wet years. The yearly mean NEE was 6.81 g C m$^{-2}$ yr$^{-1}$ during the observation
period (Fig. 4f; Tables 1 and 3). Our results agree with previous findings in dryland
ecosystems, which showed that the variability in *PPT* had significant influences on the
carbon fixation of the *Caragana microphylla* shrub-dominated ecosystem, leading it to
alternate rapidly between carbon sequestration and carbon emission (Jia et al., 2016;
Liu et al., 2016). However, the magnitude of the average annual *NEE* in the current
study was lower than those in a mixture of xerophytic shrub species (the mean *NEE*
was -77 g C m$^{-2}$ yr$^{-1}$ ); in a phreatophyte-dominated in China's Gurbantünggüt Desert
ecosystem, where the *NEE* ranged from -40 to -5 g C m$^{-2}$ yr$^{-1}$ (Liu et al., 2016); in a
*Lycium andersonii* and *Ambrosia dumosa* shrubland ecosystem, where the *NEE* was -
127 g C m$^{-2}$ yr$^{-1}$ (Jasoni et al., 2005); and in a mature semi-arid shrub ecosystem in
California (USA) dominated by *Adenostoma fasciculatum*, where *NEE* ranged from -
155 to -96 g C m$^{-2}$ yr$^{-1}$ (Luo et al., 2007). However, the carbon sequestration capacity
of the semi-fixed sandy land ecosystem was higher than that of a recovering sandy
grassland in our study region that was dominated by herbaceous species (the average
annual of *NEE* was 49 g C m$^{-2}$ yr$^{-1}$ from 2015-2018) (Niu et al., 2020). The most
plausible explanation for the difference between the two areas relates to differences in
the vegetation types. Zhang (2007) demonstrated that the carbon fixation capacity of



*Caragana microphylla* was higher than that of herbaceous and sub-shrub plants such
as *Artemisia frigida* in Horqin Sandy Land.
**4.2 Impacts of environmental condition on carbon fluxes**
Carbon fluxes are influenced by a variety of environmental factors in complicated
and interacting ways, and the main control factors change substantially across time
scales (Fu et al., 2009; Niu et al., 2010; Zhang et al., 2018a). At a seasonal scale, our
Random Forest results showed that *PPFD* and deep *SWC* ($SWC_{80}$) were the most
important environmental drivers for *GPP* and *NEE* (Fig. 5), which were both
significantly negatively related with *GPP* and strong positively related with *NEE* (Fig.
6), suggesting that light and soil water stress were limiting photosynthetic activity. As
the main energy source for plant photosynthesis, *PPFD* plays an important role in plant
carbon fixation, so with increasing *PPFD*, an ecosystem's carbon sequestration
capacity generally increases (Zhou et al., 2020; Niu et al., 2021). Our results also
demonstrated that deep *SWC* ($SWC_{80}$) affected the seasonal variation of *NEE* and *GPP*
(Fig. 5, 6), since the deep *SWC* would be closely linked to large precipitation pulses;
for example, $PPT > 20$ mm caused synchronous increases in $SWC_{80}$ in our study (Fig.
S3). This is because the larger amount of precipitation can infiltrate into the soil and
replenish the deep soil moisture, where it becomes plant-available and can sustain net
photosynthesis (Niu et al., 2020). This result was also similar with previous studies in
dryland ecosystems (Austin et al., 2004; Kurc and Small, 2007; Tang et al., 2018). For
seasonal $R_{eco}$, shallow *SWC* ($SWC_{10}$) was the most important factor, followed by deep
*SWC* ($SWC_{80}$) (Fig. 5, 6). Smaller rainfall events ($PPT < 20$ mm; Fig. S3) may alter the
shallow *SWC* and increase shallow soil microbial respiration (Thomey et al. 2011); the
duration and extent of the microbial metabolic reaction appear to be tightly linked with
the availability of shallow soil water content (Huxman et al., 2004). In addition, large
rainfall pulses ($PPT > 20$ mm; Fig. S3) trigger plant root activity in deeper soil layers
(Potts et al., 2006). These findings suggest that precipitation mainly affects carbon
fluxes (*NEE*, $R_{eco}$, and *GPP*) at a seasonal scale by affecting *SWC* in different soil layers
in our research system.





At a monthly scale, soil temperature was an essential factor that determined the
carbon fluxes, followed by water-related factors such as the monthly total $PPT$ and
$SWC_{30}$, $SWC_{80}$, and $SWC_{10}$ (Table 3; Figs. 7-9). $T_s$ and $SWC$ are often regarded as the
primary regulators of ecosystem respiration (Helbling et al., 2003; Kelsey et al., 2011;
Zhang et al., 2018b; Chang et al., 2021), and our results are consistent with this view.
$R_{eco}$ increased with increasing shallow soil temperature ($T_{s10}$), monthly total $PPT$, and
shallow $SWC$ ($SWC_{10}$) (Fig. 8). The increase was exponential between $R_{eco}$ and $T_{s10}$ (Fig.
8a), which is most likely explained by the influence of soil temperature on microbial
activity, root respiration, and soil enzyme decomposition (Jassal et al., 2008; Wang et
al., 2014). $R_{eco}$ increased significantly with linear increases in the moisture-related
factors ($PPT$ and $SWC_{10}$) (Fig. 8b, c). This may be because root activity regulates the
decomposition of soil organic matter and its influence on the microbial community can
limit or increase $R_{eco}$ (Moyano et al., 2012; Wang et al., 2014).
$GPP$ also increased with increasing $T_{s10}$, total $PPT$, $SWC_{80}$ and $SWC_{10}$ at monthly
scale (Fig. 9). $T_s$ is one of the most important environmental influences on the formation
and function of the photosynthetic apparatus (Georgieva and Yordanov, 1993; Huxman
et al., 2004; Lin et al., 2005). Water is also the main variable influencing plant
productivity and the carbon cycle in water-limited ecosystems, plant may rise their
photosynthetic rates in reaction to $PPT$ by increasing leaf-level $CO_2$ exchange, adding
more leaf area incrementally, or through a combination of both responses (Liu et al.,
2012; Hao et al., 2013; Niu et al., 2020, 2021). $PPT$ events may influence $GPP$ and $R_{eco}$
differently, thus changing the balance between them and changing the monthly $NEE$
(Hao et al., 2013). Our studies are similar to previous research: $GPP$ was more sensitive
than $R_{eco}$ to $PPT$ (Figs. 8b, 9b); the slope of the response was higher for $GPP$ (1.33)
than for $R_{eco}$ (0.92) in this area of the sandy grassland and in a sandy maize cropland
ecosystem (Niu et al., 2020, 2021).
At the annual timescale, $PPT$ was the preponderate factor that regulated the annual
$NEE$ in our semi-fixed sandy land. $NEE$ was significantly negatively correlated to $PPT$
on an annual basis during the study period (Table 3). Most previous studies showed that
the magnitude and number of $PPT$ incidents are important factors in regional climate





change, as these factors can convert biological processes at an ecosystem level (Hao et
al., 2013; Liu et al., 2012). The total *PPT* and related changes of *SWC* perform the most
important part in drylands through their impact on plant photosynthesis by altering
stomatal conductance and leaf area (Harper et al., 2005; Ford et al., 2008; Niu et al.,
2020). However, they also alter ecosystem respiration processes by affecting substrate
availability of soil microbial respiration (Epstein et al., 1997; Hao et al., 2013; Shi et
al., 2014; Niu et al., 2020).

In summary, the three carbon fluxes (*NEE*, $R_{eco}$, and *GPP*) are not affected by

single factors, but rather by a combination of a variety of environment parameters.
However, when the time scale gets longer, the important factors affecting the changes
of *NEE*, $R_{eco}$, and *GPP* preferred to converge. At the daily timescale, their values were
influenced by radiation, temperature, and water, but at the monthly and annual
timescale, the primary governing factor varied to water. Generally, water performed a
key role in the change of ecosystem carbon fluxes at all the time scale.
**5. Conclusion**

We studied the carbon fluxes and their environmental driving factors at different time

scales in a semi-fixed sandy land. Our results indicated that the carbon source or sink
intensity of the ecosystem, which is undergoing restoration to combat desertification in
the Horqin Sandy Land, and it's consistent with our hypothesis, is greatly uncertain due
to the complex and interacting influences of environmental factors, especially for
precipitation. In the wet years (2019 and 2021), the semi-fixed sandy land was a carbon
sink, whereas in the dry years (2017 and 2020) and the normal year (2018), the system
showed a carbon source, with a mean annual *NEE* of 6.81 g C $m^{-2}$ $yr^{-1}$.

We determined the primary governing factors of *NEE*, $R_{eco}$, and *GPP* using

correlation analyses, Random Forest models, and multiple stepwise regression analysis.
*PPFD* and deep *SWC* ($SWC_{80}$) were important drivers for the seasonal variation of *NEE*
and *GPP*, whereas both shallow and deep *SWC* ($SWC_{10}$ and $SWC_{80}$) were important
drivers for $R_{eco}$. At a monthly scale, $T_s$, *PPT*, and $SWC_{10}$ were strong positively related
to $R_{eco}$ and *GPP*, whereas *NEE* was strong negatively related to $T_{s50}$, *PPT*, and $SWC_{30}$.
Annual *NEE* was strongly correlated with the total *PPT*. Water performed a key role in



the changes of ecosystem carbon fluxes in our semi-fixed sandy land ecosystem. If
regional precipitation increases in the future, the potential carbon sequestration in semi-
fixed sandy land ecosystem is likely to increase.
**Data Availability**
In agreement with the FAIR Data standards, the data used in this article are archived,
published, and available in a dedicated repository: https://doi.org/10.4121/20071877.
**Author contributions**
YQL, YYN, WL, XYW, and YC designed the study, YYN analyzed the data. YYN
drafted the manuscript. All co-authors had a chance to review the manuscript and
contributed to discussion and interpretation of the data.
**Competing interests**
The authors declare that they have no known competing financial interests or personal
relationships that could have appeared to influence the work reported in this paper.
**Acknowledgments**
This research was supported by the National Natural Science Foundation of China
(grants 31971466 and 32001214) and the National Key Research and Development
Program of China (2017YFA0604803 and 2017YFA0604801).

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

**Figure captions**
**Fig. 1.** Locations of study area. (a) and (b) are photos of observation site during the
growing and dormant seasons, respectively.
**Fig. 2.** (a) Seasonal dynamics of the daily average vapor pressure deficit (*VPD*), (b)
photosynthetic photon flux density (*PPFD*), (c) air temperature ($T_a$), (d) the soil
temperature ($T_s$) and (e) soil water content (*SWC*) at depths of 10, 30, 50, and 80 cm,
and (f) daily precipitation (*PPT*).
**Fig. 3.** Seasonal variability of the daily mean carbon fluxes (*NEE*: net ecosystem $CO_2$
exchange, $R_{eco}$: ecosystem respiration, and *GPP*: gross primary productivity) from 2017
to 2021.
**Fig. 4.** (a-e) Monthly and (f) inter-annual variations of the cumulative carbon fluxes
(*NEE*: net ecosystem $CO_2$ exchange, $R_{eco}$: ecosystem respiration, and *GPP*: gross
primary productivity) from 2017 to 2021.
**Fig. 5.** Important values of the Random Forest model analysis for the carbon flux (*NEE*:
$CO_2$ net ecosystem exchange; $R_{eco}$: ecosystem respiration; *GPP*: gross primary
productivity) during the 2017 to 2021 growing seasons. ** and * refer to significance
at 0.01 and 0.05 levels, respectively. Variables: *PPFD*: mean photosynthetic photon
flux density; $T_a$: mean air temperature; *VPD*: mean vapor pressure deficit; *PPT*: daily
total precipitation; $T_s$ and *SWC*: mean soil water content at depths of 10, 30, 50, and 80
cm.
**Fig. 6.** Relationships between seasonal net ecosystem carbon exchange (*NEE*), gross
primary productivity (*GPP*), and ecosystem respiration ($R_{eco}$) and the (a) mean
photosynthetic photon flux density (*PPFD*) and (b-d) mean soil water contents at depths
of 10, 30, and 80 cm ($SWC_{10}$, $SWC_{30}$, and $SWC_{80}$, respectively).
**Fig. 7.** Relationships between monthly net ecosystem carbon exchange (*NEE*) and the
main meteorological factors: soil temperature at a depth of 50 cm ($T_{s50}$), mean soil water
content at a depth of 30 cm ($SWC_{30}$), vapor-pressure deficit (*VPD*), and photosynthetic
photon flux density (*PPFD*).





**Fig. 8.** Relationships between monthly ecosystem respiration ($R_{eco}$) and the main
environmental factors: the soil temperature at a depth of 10 cm ($T_{s10}$), the total
precipitation ($PPT$), and the soil water content at a depth of 10 cm ($SWC_{10}$).
**Fig. 9.** Relationships between monthly gross primary productivity ($GPP$) and the main
environmental factors: soil temperature at a depth of 10 cm ($T_{s10}$), total precipitation
($PPT$), and soil water content at depths of 80 cm ($SWC_{80}$) and 10 cm ($SWC_{10}$)


**Fig. 1.**

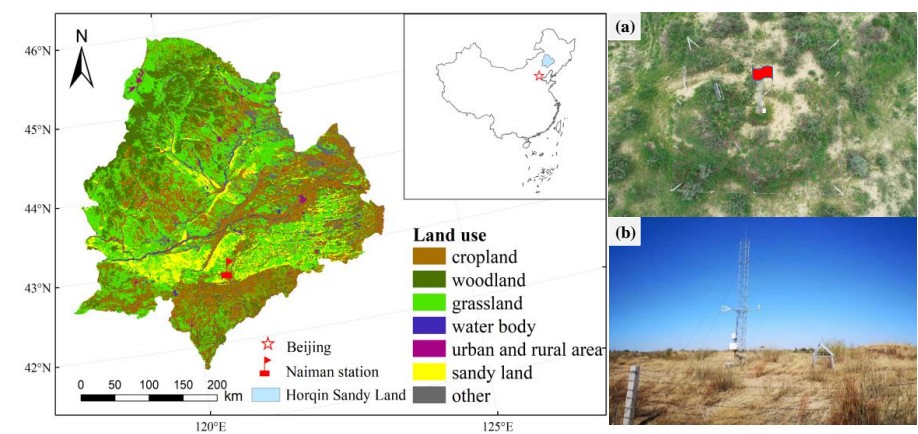


**Fig. 2.**

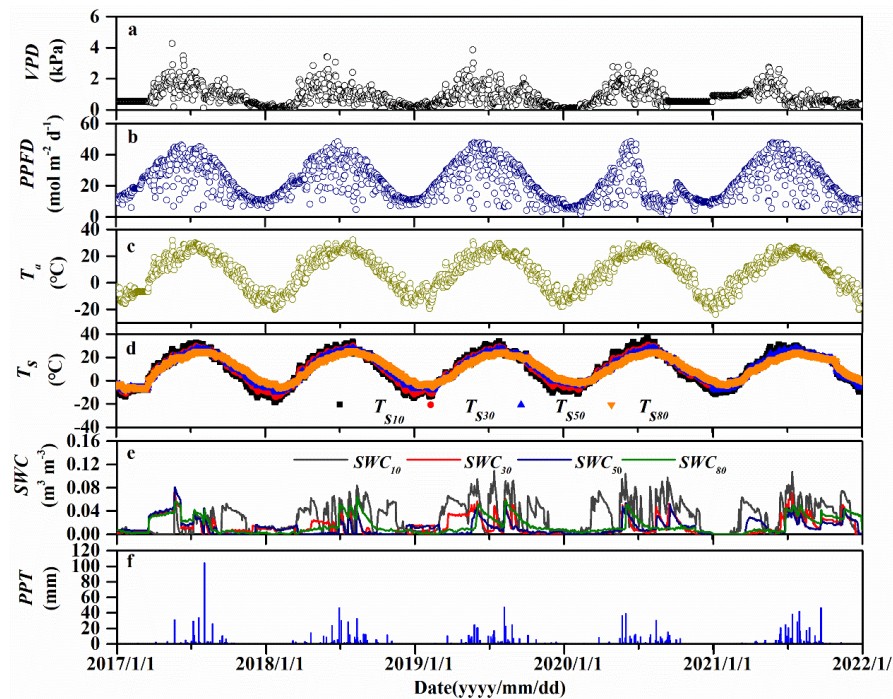




**Fig. 3.**

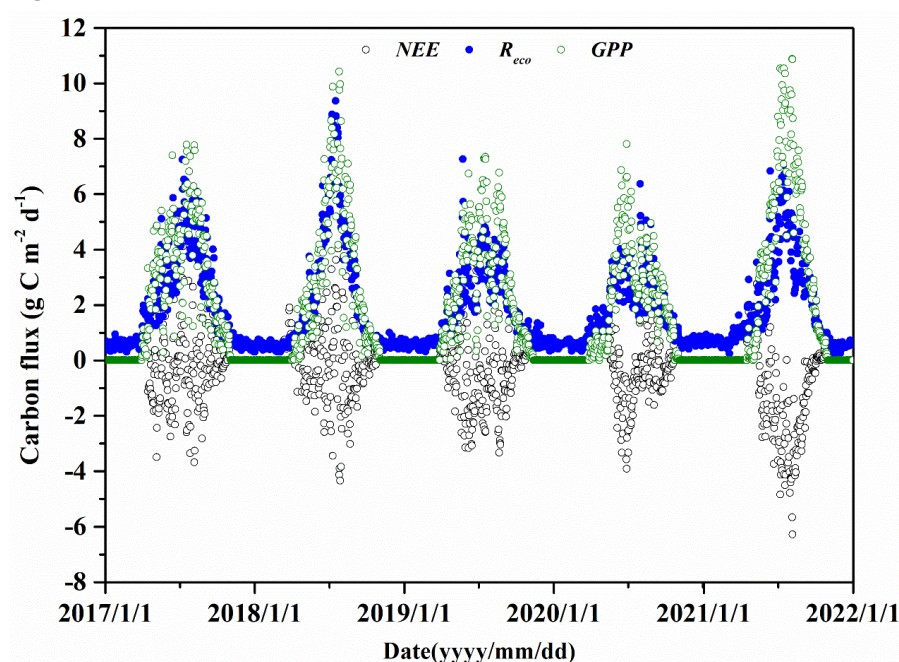


**Fig. 4.**

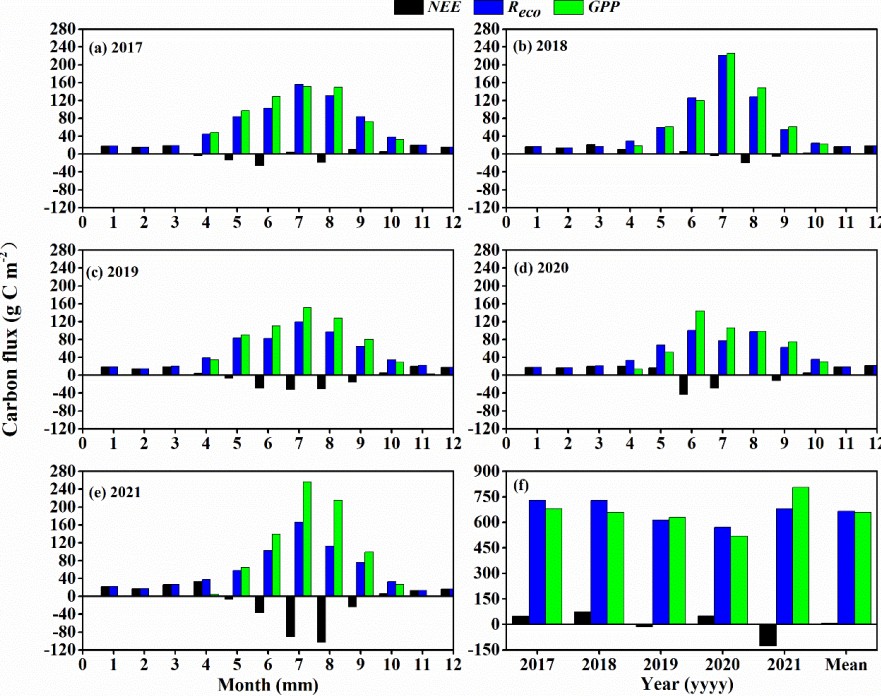




**Fig. 5.**

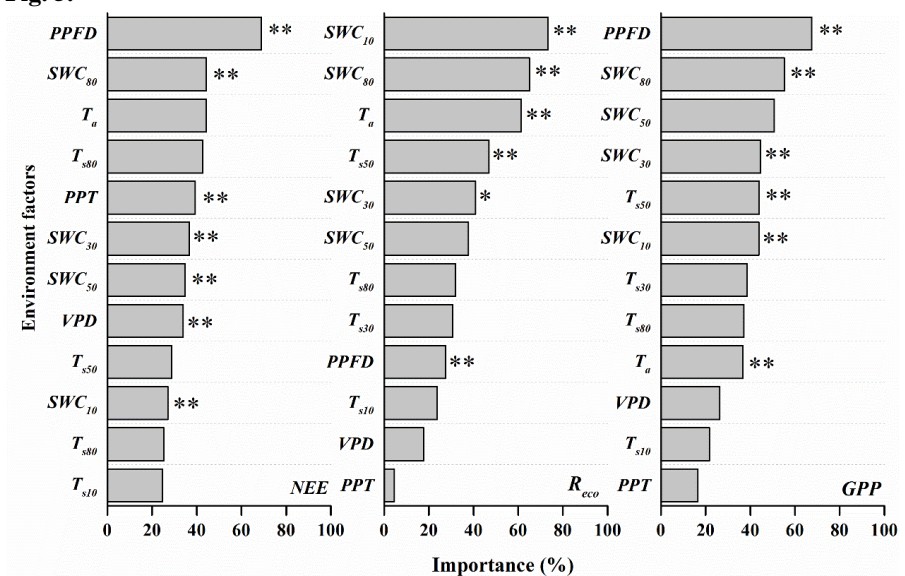


**Fig. 6.**

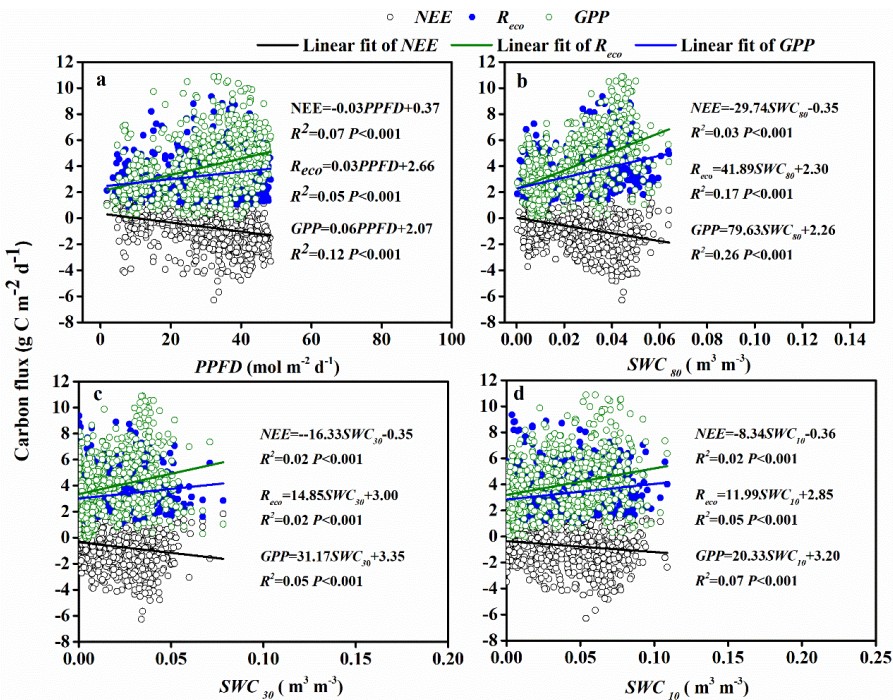






**Fig. 7.**

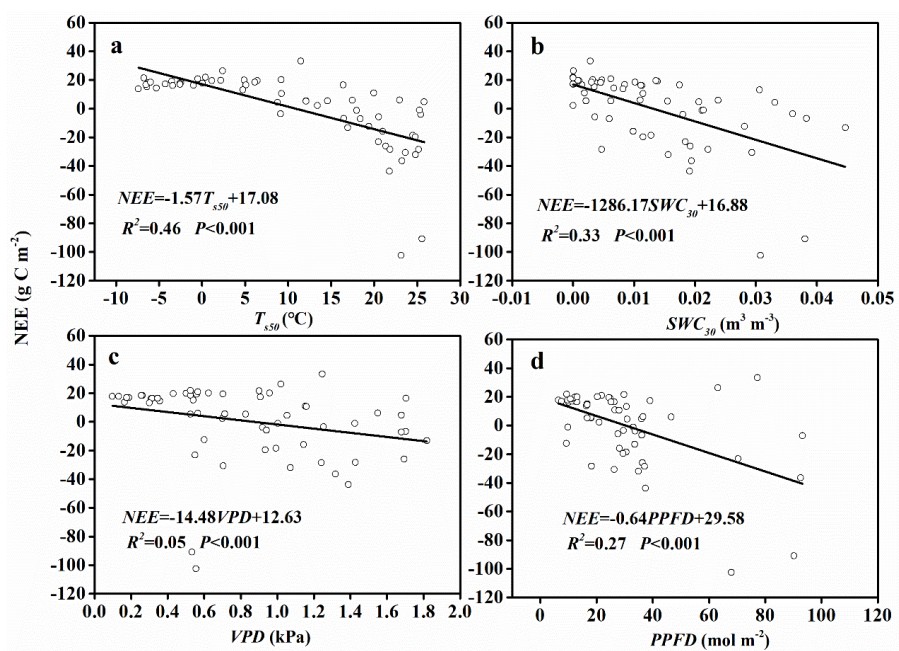


**Fig. 8.**

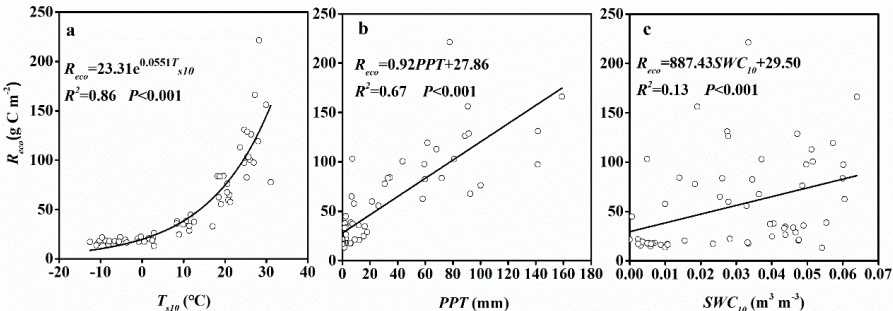






**Fig. 9.**

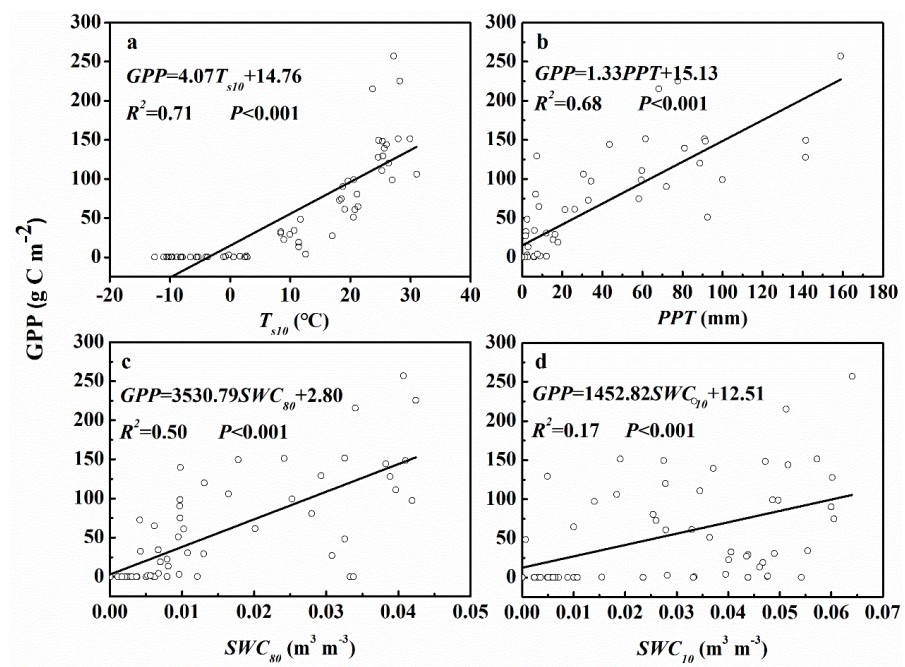







**Tables**
**Table 1** Annual mean meteorological factors from 2017 to 2021 in the semi-fixed sandy
ecosystem. Values of a variable labeled the same letter did not differ significantly
among the years.

| Year | $T_a$ | VPD | PPFD | PPT | $SWC_{10}$ | $SWC_{30}$ | $SWC_{50}$ | $SWC_{80}$ | $T_{s10}$ | $T_{s30}$ | $T_{s50}$ | $T_{s80}$ |
|------|-------|-----|------|-----|-----------|-----------|-----------|-----------|-----------|-----------|-----------|-----------|
| 2017 | 7.90a | 1.00b | 24.00b | 312.80b | 0.014a | 0.015b | 0.016c | 0.016b | 8.79a | 9.16a | 9.68a | 10.28a |
| 2018 | 8.03a | 0.76a | 23.55b | 350.80c | 0.028b | 0.009a | 0.009a | 0.013b | 9.08a | 9.61a | 10.32a | 10.88a |
| 2019 | 8.48a | 0.79a | 23.86b | 382.20d | 0.036c | 0.015b | 0.015b | 0.016b | 9.67a | 10.11a | 10.71a | 11.04a |
| 2020 | 8.26a | 0.79a | 15.44a | 312.00a | 0.034c | 0.009a | 0.009b | 0.011a | 10.83a | 11.02a | 11.11a | 11.20a |
| 2021 | 8.08a | 0.83a | 23.87b | 430.404 | 0.034c | 0.015b | 0.015c | 0.019c | 11.34a | 11.40a | 11.33a | 11.45a |
| Mean | 8.15a | 0.83a | 22.14b | 357.64 | 0.029 | 0.013 | 0.011 | 0.015 | 9.94a | 10.26a | 10.63a | 10.97a |

Note: $T_a$ (℃): air temperature; *VPD* (kPa): vapor pressure deficit; *PPFD* (mol m$^{-2}$ d$^{-1}$):
photosynthetic photon flux density; *PPT* (mm): total precipitation; *SWC* (m$^3$ m$^{-3}$) and
$T_s$ (℃): soil water content and soil temperature at depths of 10, 30, 50, and 80 cm,
respectively.




**Table 2** The results of multiple stepwise regression analysis of carbon fluxes (*NEE*: net
ecosystem exchange; $R_{eco}$: ecosystem respiration; *GPP*: gross primary production)
against the potential drivers during the growing season from 2017 to 2021. All
regressions were statistically significant at $P < 0.001$.

| Stepwise regression equation | $F$ | $R^2$ |
|---|---|---|
| $NEE = -0.69\ T_{s50}+3.15$ | 51.70 | 0.46 |
| $NEE = -0.53\ T_{s50}+0.33SWC_{30}+3.52$ | 35.10 | 0.54 |
| $NEE = -0.72\ T_{s50}+0.35\ SWC_{30}+0.32\ VPD+4.54$ | 29.57 | 0.59 |
| $NEE = -0.65\ T_{s50}+0.31\ SWC_{30}+0.39\ VPD+0.29\ PPFD+4.43$ | 28.75 | 0.65 |
| $R_{eco} = 0.84\ T_{s10}+4.09$ | 134.62 | 0.70 |
| $R_{eco} = 0.51\ T_{s10}+0.45\ PPT+3.47$ | 113.46 | 0.80 |
| $R_{eco} = 0.61\ T_{s10}+0.50\ PPT-0.22\ SWC_{10}+4.59$ | 91.96 | 0.83 |
| $GPP = 0.34\ T_{s10}+5.71$ | 144.51 | 0.71 |
| $GPP = 0.39\ T_{s10}+0.13\ PPT+4.87$ | 124.75 | 0.81 |
| $GPP = 0.40\ T_{s10}+0.12\ PPT+340.95\ SWC_{80}+5.18$ | 103.03 | 0.84 |
| $GPP=0.42T_{s10}+0.12PPT+333.20SWC_{80}+214.28SWC_{10}+6.82$ | 82.62 | 0.85 |

Note: $T_a$ (°C): air temperature; *VPD* (kPa): vapor pressure deficit; *PPFD* (mol m$^{-2}$ d$^{-1}$):
photosynthetic photon flux density; *PPT* (mm): total precipitation; *SWC* (m$^3$ m$^{-3}$) and
$T_s$ (°C): soil water content and soil temperature at depths of 10, 30, 50, and 80 cm,
respectively.



**Table 3** The results of correlation analysis (Pearson's $r$) between the inter-annual
carbon fluxes ($NEE$: net ecosystem exchange; $R_{eco}$: ecosystem respiration; and $GPP$:
gross primary production) and the potential drivers from 2017 to 2021. Significance: *,
$P < 0.05$.

| Carbon flux | $PPT$ | $VPD$ | $PPFD$ | $T_a$ | $T_{s10}$ | $T_{s30}$ | $T_{s50}$ | $T_{s80}$ | $SWC_{10}$ | $SWC_{30}$ | $SWC_{50}$ | $SWC_{80}$ |
|---|---|---|---|---|---|---|---|---|---|---|---|---|
| $NEE$ | -0.89* | 0.06 | -0.32 | -0.14 | -0.69 | -0.67 | -0.63 | -0.63 | -0.43 | -0.65 | -0.46 | -0.80 |
| $R_{eco}$ | 0.06 | 0.46 | 0.74 | -0.82 | -0.57 | -0.60 | -0.65 | -0.55 | -0.69 | 0.12 | 0.16 | 0.45 |
| $GPP$ | 0.74 | 0.26 | 0.76* | -0.45 | 0.15 | 0.11 | 0.05 | 0.12 | -0.13 | 0.58 | 0.45 | 0.93* |

Note: $T_a$ (°C): air temperature; $VPD$ (kPa): vapor pressure deficit; $PPFD$ (mol m$^{-2}$ d$^{-1}$):
photosynthetic photon flux density; $PPT$ (mm): total precipitation; $SWC$ (m$^3$ m$^{-3}$) and
$T_s$ (°C): soil water content and soil temperature at depths of 10, 30, 50, and 80 cm,
respectively.