# Peer review of "Variations of carbon flux at different time scales in a semi-fixed sandy"

_Biogeosciences, 2022_

## Author Comment (AC1)

30 January 2023

Dear Editor:

Thank you for coordinating the review of our manuscript Variations of carbon flux at different time scales in a semi-fixed sandy land ecosystem in Horqin Sandy Land, China (**bg-2022-171**). In the rest of this letter, we have provided details of our responses to the review comments. We hope that these responses and the resulting changes will be acceptable, but we will be happy to work with you to resolve any remaining issues.

Sincerely,

Yuqiang Li (on behalf of all authors)

Reviewer 1

*Biogeosciences*

RE: Submission of the revised manuscript (No. **bg-2022-171**): Variations of carbon flux at different time scales in a semi-fixed sandy land ecosystem in Horqin Sandy Land, China

Dear Reviewer#1:

   Thank you for your assistance in the review of our manuscript. We have revised the manuscript carefully according to your comments.

Our detailed responses to comments are presented in the remainder of this letter. All of revisions have been highlighted in red in the revised manuscript.

General comments:

I have carefully reviewed the manuscript of Niu et al on sandy land carbon fluxes and climate variability. The manuscript presents a five-year time series of standard eddy covariance carbon flux data (measured NEE with GPP and Reco estimated by accepted methods) and associated weather and soil profile data. The main novelty of the paper is the sandy land ecosystem, a degraded state of the Inner Mongolian grasslands produced by a combination of climate and land use factors. The manuscript presents a generalized exploration of the data and does not test any specific hypotheses.

We have added our two research hypotheses in the Introduction (Lines 135-137, 140-145 in the revision) and test the hypotheses (Lines 245-271, 307-317 in the revision).

The key conclusions of the paper, summarized in the latter half of the abstract, mostly repeat ideas that are well-established in semiarid ecosystem flux literature, such as the idea that many dryland ecosystems have mean annual NEE~0 but pivot between carbon sinks/sources in wet/dry years. Or that precipitation, temperature and soil water content are "dominant controls."

We have revised the Abstract to clarify the novelty of our study. First, there has been little research in our study area, which is an important ecological area of China. Second, our study was conducted at multiple time scales and thereby revealed changes in the dominant factors that affect *NEE* in response to changes in the time scale. We emphasized carbon flux changes in semi-fixed sandy land ecosystems during the period of recovery from severe desertification, which is a period that has received insufficient attention in the literature (115-121 in the revision).

In the end, I find myself unable to clearly answer the important referee question: "Do the results support the key points/conclusions?" because the results are relatively unstructured, and the key points are overly general. The presentation of the manuscript follows accepted guidelines for organization and formatting.

We have added our two research hypotheses in the Introduction (Lines 135-137, 140-145 in the revision) and have restructured the results to clarify how our results relate to the two research hypotheses and the impact of our study. In particular, we have strongly supported the importance of water availability during the growing season and changes in the impact of the environmental factors on carbon fluxes with changing time scales (Lines 245-271, 307-317, Fig. 3, Fig. 6, and Table 3 in the revision).

The writing contains numerous faults in English language usage, although I was able to follow the meaning in nearly all cases.

We have asked Geoffrey Hart (ghart@videotron.ca/geoff@geoff-hart.com), an English science editor with more than 35 years of experience, to ensure that the quality of the language will be acceptable. Please contact him if necessary to confirm that he has performed this work or if you have any questions about the nature of the work that he has done.

Figures are of medium to poor quality, both in terms of visual presentation (especially readability of high-frequency data) and of conceptual representation which is an indicator of the carbon sink or source in terrestrial ecosystems. Therefore, comprehending the dynamics processes and underlying mechanisms of NEE is a crucial issue in global change research.

We have added graphs of the variation of the carbon fluxes and their relationships with precipitation and soil water content in dry and wet years during the growing season (Fig. 3a-e in the revision), and have also added the relationship between monthly-scale environment factors and carbon fluxes (Fig. 6 in the revision).

Overall, my impression is that this manuscript is not suitable for publication due to a lack of clear, testable hypotheses (or clear goals). This deficit leads to other "downstream" problems including poorly-conceived figures. The literature cited is missing many major bodies of work from several other continents. I encourage the coauthors to examine the literature more extensively, identify key knowledge gaps that could be addressed with this dataset, and write a manuscript that leverages these

valuable data to advance understanding. Given the large number of single-site, multi year eddy covariance studies published over the last 25 years, some suggested avenues for enhancing novelty include 1) comparing/contrasting the sandy land flux behavior with other dryland sites. 1) For example, it appears that the mean annual precipitation and seasonal distribution of this site are quite similar to grassland sites in New Mexico and possibly Arizona, USA, but with very different temperatures; 2) Showing how these flux data change the (or do not change) what is presently assumed about fluxes from such regions using currently available tools (often ecosystem models and/or remote sensing).

We have added the two specific research hypotheses we used (Lines 135-137, 140-145 in the revision), and have compared the changes at our site with values from 5 previous studies (Lines 313-337 in the revision), and have added a discussion of the impacts of the changes of the environmental factors (precipitation, temperature, and solar radiation) on the carbon fluxes (Lines 339-352 in the revision). In terms of the literature review, we have cited 17 additional recent studies.

Although your suggestions for improving the novelty of our study are interesting, we note that novelty is not the only goal of research and not even the most important goal. Replication of previous research in under-studied regions is also a valid goal, and was the purpose of our approach. However, we have clarified the novelty of our research in the Abstract (see our response earlier in this letter). We have also added these novel aspects in the Introduction (115-121 in the revision).

Below are some specific comments included in the hope that they may be useful for advancing this work.

1.21: What is semi-fixed sandy land?
We have added a definition of semi-fixed sand (Lines 78-80 in the revision). In summary, this is based on the soil texture (sandy) and the vegetation cover.
Zhao, H. L., Zhao, R. L., Zhao, X. Y., Zhang, T. H.: Ground discriminance on positive and negative processes of land desertification in Horqin Sand Land (in Chinese), J. of Desert Research, 28, 8-15. http://210.72.80.159/jweb_zgsm/EN/Y2008/V28/I1/8, 2008.

2. 25-29: This part would be stronger if it were framed around hypothesis testing. Instead, the writing focuses us on a few statistical techniques for assessing variable importance. What new knowledge is produced by this study that is transferable to other times and places?

We have revised this description to clarify that we examined the effects of seasonal-scale and inter-annual values of the environmental factors on carbon fluxes (Lines 285-327 in the revision). We have added our two research hypotheses (Lines 135-137, 140-145 in the revision) and restructured the results to support these hypotheses (Lines 245-271, 307-317, Fig. 3, Fig. 6, and Table 3 in the revision).

3. 29-32: This is all well-known from many other dryland studies.

We have deleted the well-known information in the revision.

4. Since the mean annual NEE = zero (within the uncertainties of the method), there could be more interest in evaluating the magnitude of the deviations from the mean (e.g. source and sink function in dry and wet years).

Because *NEE* responded differently in different years, relying on the overall average could be misleading. Although we have retained that data, we have clarified that the important differences in our study were between the wet and dry years (Lines 22-26, 460-463 in the revision).

5. 45: As a result of what? The prior sentence talks about how NEE is mathematically related to GPP and Reco, but it does not logically follow that these relationships establish a reason that understanding NEE is crucial.

We have clarified that *NEE* is crucial because it reflects the balance between photosynthesis and respiration and thus, affects whether a site will be a carbon source or sink (Lines 44-47 in the revision).

6. 62: This is a potentially promising topic for development of new knowledge, if the manuscript focuses on testing controlling mechanisms for sandy lands, especially if there are different controls in sandy lands than for other semiarid ecosystems.

We have clarified that our study focused on the control mechanisms at a desertified sandy site that is recovering from degradation. Our goal was not to compare our results with other sites during our study (to do so, we would have needed to add a second site in our analysis), but we do perform that comparison in the Discussion, were we have added citations of 5 studies to provide a comparison (Lines 339-352 in the revision).

7. 72; please define semi-fixed sands.
We have defined this term in Lines 78-80 in the revision.
Zhao, H. L., Zhao, R. L., Zhao, X. Y., Zhang, T. H.: Ground discriminance on positive and negative processes of land desertification in Horqin Sand Land (in Chinese), J. of Desert Research, 28, 8-15. http://210.72.80.159/jweb_zgsm/EN/Y2008/V28/I1/8, 2008.

8. 74: was a carbon source

We have changed this to "carbon source" in the revision (Line 83)."

9. 113: I am trying to determine what to expect that is new and different in this study as compared to the extensive body of literature in other semiarid ecosystems and as compared to the works cited here of Niu (2020, 2021) in sandy lands. Can this be made more clear?

We have clarified that the semi-fixed sandy land at our study site is recovering after severe ecosystem degradation leading to desertification, which is different from most previous studies of other semiarid ecosystems (Lines 62-66, 70-74, 114-132 in the revision). As compared to the works cited here of Niu (2020, 2021) in sandy lands, the main difference is the vegetation and land-use types (Lines 149-152 in the revision).

10. 169: I find these paragraphs very difficult to read, since they are packed with numbers and generally lacking any narrative thread to tie them together. It could be more effective to report means and ranges in a table and then use the text to illustrate key features of the data (features related to testing the study hypotheses, such as extreme seasons/years)

We have removed the redundant data and only show the most important data that are most relevant to the study. We have revised the description related to Figure 2 and Table 2 (lines 223-243 in the revision).

11. 170: to my understanding, describing a solar-cycle time series as a unimodal distribution is not appropriate, because a time series (which may be unimodal) is a sequence, not a distribution of (random) samples from a population.

We have changed our description to "unimodal trends", since we show the trends over time rather than a statistical distribution graph (Line 224 in the revision).

12. 190: VPD is an example of a mechanistic control with potential to be explored. Is there a hypothesis related to VPD? See, for example, recent papers by Kim Novick on this topic and specifically the inferences related to drylands with respect to canopy flux controls by VPD vs. SWC.

We have carefully read the literature that you recommended and analyzed how *VPD* influences the carbon fluxes. Because *VPD* was not a major factor that affected the carbon fluxes in our study ecosystem, we did not add a discussion of its mechanisms and have instead retained the original description (Lines 239-243 in the revision).

13. 194: this paragraph is mostly generalizations, such as "showed obvious seasonal changes."   It would be better to describe what these dynamics are. Better yet, refer to Figure 3 for the dynamics and use the text to point out key features related to the study questions.

We have added more specific descriptions of the seasonal changes (Lines 245-250 in the revision).

14. 216: it would be more compelling to lead with the study hypotheses and then use the statistical techniques to test the hypotheses.

We have added our two research hypotheses (Lines 135-137, 140-145 in the revision) and restructured the results to support our hypotheses (Lines 245-271, 307-317, Fig. 3, Fig. 6, and Table 3 in the revision).

15. Figure 2 and Figure 3: While the figures contain valuable information, the high frequency nature, many time series, many panels, and 5-year coverage period makes it difficult to discern anything meaningful beyond what we could learn from a table of climate variables. If there key study hypotheses being tested that depend upon these variables (e.g. wet vs. dry years, or SWC variability at differing depths), the figures

should highlight the key aspects of these datasets.

We included Figure 2 to show the changes of meteorological factors throughout the study period to reveal important environmental similarities and differences between years (i.e., factors that do or don't explain differences in the carbon fluxes). However, we have added Figure 3 to show the changes of the three carbon fluxes during the same period and their relationships with precipitation and soil water content during the growing season in representative dry and wet years to clarify these relationships. We also added an analysis of the relationships between the monthly-scale environmental factors and the carbon fluxes (Fig, 6 and Table 3 in the revision).

16. Figure 4: This figure with monthly values could potentially provide some interesting ideas for hypotheses to test. For example, the large GPP and carbon sink NEE in summer 2021 might inspire a figure or other analysis relating monthly fluxes to key hypothesized controls.

Based on the monthly-scale values of the three carbon fluxes in Figure 4, we have added Figure 6 and Table 3 to describe the relationships between the monthly-scale environmental factors and the three carbon fluxes during the growing season, and we discuss how these environmental factors affected $GPP$, $NEE$ and $R_{eco}$ (Lines 356-368 in the revision).

17. Figure 6 strikes me as rather oversimplified, and I struggle to learn anything new from this figure. The fits are generally poor, and the time scales might not even make sense. For example, a scatterplot of high-frequency data like this does not account for the Birch Effect, in which the largest respiration values in dryland ecosystems often follow (lag) by several days, pulses in SWC. Graphical tests of key drivers (e.g. GPP vs. SWC at 3 depths) might be more tractable at the monthly scale, as suggested by the interesting results for summer 2021 in Figure 4.

We have deleted the high-frequency data at a daily scale and have replaced them with the relationship between the monthly-scale environmental factors and the three carbon fluxes during the growing season (Fig. 6 and Table 3 in the revision).

18. Figure 7-9: This shows some potentially interesting results, but there are problems with this figure. First, it does not follow from any hypotheses posed in the introduction about the key controls. Second, lines are fit to data that are apparently nonlinear. For example, panel a shows a flat response of NEE to Ts50 between -8C and +8C, and then a declining relationship, and there is likely ecological meaning in that nonlinear patterns. Second, these simple scatter plots to not seem to account for the seasonality of the ecosystem. In other words, we are presented the relationship between SWC and NEE in all months of the year, when we have major reasons to expect different relationships within the growing season or outside it. One suggestion for improving this analysis would be to present a bivariate relationship (like these ones) that is screened for a given range of values in other variables. For example, plot GPP vs. SWC for Ts50 between 5-10 C (or whatever might make sense). Or plot GPP vs. SWC for summer growing

season only.

We have replaced Figures 7 to 9 in the original manuscript with the relationships between the monthly-scale environmental factors and the three carbon fluxes during the growing season (Fig. 6 and Table 3 in the revision). In addition, we used nonlinear regression for some of the graphs in Figure 6 where that provided a better fit. We then discuss these relationships in terms of the research hypothesis. In summary, the effects of drought during the growing season were most important, and this was revealed in the long-term (annual) time scale (Lines 245-271, 307-327 in the revision).

19. Table 1: this is a valuable table. Please express significant digits such that they represent the true uncertainty. For example, Uncertainty in precipitation measurements with a tipping bucket gauge can range from 5-15% or more. Therefore, P = 312.80 mm, which communicates precision to 1x10-5 m, is not reasonable. Perhaps 312 or 310 or 300 would be more representative. Please consider this for each variable shown.

As you note, providing multiple decimal places of precision may not be appropriate. We have therefore presented the precipitation values as integers (Lines 233-234, Table 2 in the revision). Calculating the optimal precision (integers versus values expressed to the nearest 2 or 10 mm) would be beyond the scope of our study.

Suggested reading:

In addition to the valuable works cited herein, mainly focused on China sites, there exists extensive literature on semiarid fluxes by the eddy covariance method across multiple other continents. Please see for example the following short list and the references in these papers:

Novick, Kimberly A., et al. "The increasing importance of atmospheric demand for ecosystem water and carbon fluxes." Nature climate change 6.11 (2016): 1023-1027.

Haverd, Vanessa, et al. "Carbon cycle responses of semiâ□□arid ecosystems to positive asymmetry in rainfall." Global change biology 23.2 (2017): 793-800.

Scott, Russell L., et al. "The carbon balance pivot point of southwestern US semiarid ecosystems: Insights from the 21st century drought." Journal of Geophysical Research: Biogeosciences 120.12 (2015): 2612-2624.

Biederman, Joel A., et al. "Terrestrial carbon balance in a drier world: the effects of water availability in southwestern North America." Global change biology 22.5 (2016): 1867-1879.

Biederman, J. A., Scott, R. L., Arnone III, J. A., Jasoni, R. L., Litvak, M. E., Moreo, M. T., ... & Vivoni, E. R. (2018). Shrubland carbon sink depends upon winter water availability in the warm deserts of North America. Agricultural and Forest Meteorology, 249, 407-419.

Dannenberg, Matthew P., et al. "Exceptional heat and atmospheric dryness amplified losses of primary production during the 2020 US Southwest hot drought." Global change biology (2022).

Scott, Russell L., et al. "Commonalities of carbon dioxide exchange in semiarid regions with monsoon and Mediterranean climates." Journal of arid environments 84 (2012): 71-79.

Castellanos, Alejandro E., et al. "Plant functional diversity influences water and carbon fluxes and their use efficiencies in native and disturbed dryland ecosystems." Ecohydrology: e2415.

We have carefully read the literature that you recommended and have cited 4 of the 8 references you suggested in the revision.

Thank you for your efforts to improve our paper. We hope that our responses and the resulting changes will be acceptable, but we will be happy to work with you to resolve any remaining issues.

Sincerely,

Yuqiang Li, Ph.D.

Northwest Institute of Eco-Environment and Resources

Chinese Academy of Sciences

320 Donggang West Road, Lanzhou, 730000, China

Phone/Fax: 86-931-496-7219

E-mail: liyq@lzb.ac.cn

---

## Author Comment (AC4)

30 January 2023

Dear Editor:

Thank you for coordinating the review of our manuscript Variations of carbon flux at different time scales in a semi-fixed sandy land ecosystem in Horqin Sandy Land, China (**bg-2022-171**). In the rest of this letter, we have provided details of our responses to the review comments. We hope that these responses and the resulting changes will be acceptable, but we will be happy to work with you to resolve any remaining issues.

Sincerely,

Yuqiang Li (on behalf of all authors)

Reviewer 2

*Biogeosciences*

RE: Submission of the revised manuscript (No. **bg-2022-171**): Variations of carbon flux at different time scales in a semi-fixed sandy land ecosystem in Horqin Sandy Land, China

Dear Reviewer#2:

Thank you very much for your assistance in the review of our manuscript. We have revised the manuscript carefully according to your comments.

Our detailed responses to comments are presented in the remainder of this letter. All of the revisions have been highlighted in red in the revision.

This study used the eddy covariance method to assess carbon fluxes over five years (2017-2021) at a semi-arid, temperate shrubland in China. The site was severely desertified due to overgrazing but has undergone restoration through grazing exclosures. The aim of the study was to analyse what environmental variables affect carbon fluxes over different time scales, from monthly to interannual, and precipitation and soil water content was hypothesized to have a dominant effect on carbon fluxes. The methods are poorly described, with authors referring to other published studies rather than thoroughly describing the equipment at their study site and their statistical analysis.

We have added more detailed descriptions of the key methods in the revision, including details of the Random Forest analysis (Lines 190-210 in the revision), and the statistical analyses (Lines 211-220 in the revision). In addition, Table 1 provides details of the measurement instruments we used and their position relative to the soil surface.

The figures and tables are nicely done, although a few corrections are needed here too. While not highly novel, as a single site eddy covariance study, it is nevertheless a valuable contribution to our understanding of ecosystem functioning and carbon fluxes in an understudied region.

Thank you for recognizing the purpose and importance of our study. In response to reviewer 1, we have added Figure 3 to show the carbon fluxes and their relationships with precipitation and soil water content during the growing season for representative dry and wet years. We have also added Figure 6 and Table 3 to show the relationships between the monthly-scale environmental factors and the carbon fluxes.

The title is rather generic, however, so I recommend the authors selecting a more informative title.

We have revised the title as "Effects of environment factors on the carbon fluxes of semi-fixed sandy land recovering from degradation".

The site was a carbon sink in wet years (-14, -126 g C m-2 yr-1) but a carbon source in dry or average years (49-75 g C m-2 yr-1). The authors confirm their hypothesis that precipitation is an important driver of carbon fluxes and focus on this in the Conclusion and Abstract, although air and soil temperature are also important (in fact, more important for ecosystem respiration and GPP, Fig. 5) but largely neglected.

In response to reviewer 1, we have more clearly indicated our research hypotheses in the Introduction and have stated whether we confirmed our research hypotheses in the Conclusion (Lines 135-137, 140-145, 451-469 in the revision). We have added more details of our analysis of the impacts of the air and soil temperatures and of *PPFD* on the carbon fluxes (Lines 307-317, 406-430 in the revision).

The manuscript falls short of it's potential, as the authors do not discuss how (1) desertification and/or restoration or (2) warming has affected (or could have, or potentially will affect) the ecosystem. Perhaps this is because they do not know, given that there is only one site with five years of data, but this should at least be discussed, given that desertification/restoration is mentioned in both the Introduction and Conclusion. The potential effects of climate change, including both precipitation and warming, would make a valuable addition to their Conclusions, which currently only repeat their results.

We have added descriptions of how desertification may have affected the ecosystem in the Introduction (Lines 70-74, 114-132 in the revision) and Conclusion (Lines 451-463 in the revision), and have also added a note about the predicted warming trend and increased precipitation in the future, as well as how this is likely to affect the ecosystem (Lines 425-430, 465-469 in the revision).

This manuscript should be edited for proper English grammar and language throughout.

We have asked Geoffrey Hart (ghart@videotron.ca/geoff@geoff-hart.com), an English science editor with more than 35 years of experience, to ensure that the quality of the language will be acceptable. Please contact him if necessary to confirm that he has performed this work or if you have any questions about the nature of the work that he has done.

Please see specific comments below:

1.Abstract

Line 16: "Sandy land" is a regionally specific community type and not widely understood, so a better description of the specific study location is needed here in the abstract (e.g., is it arid or semi-arid? Is it a grassland or shrubland?).

We have added a definition of sandy land and a more detailed description of the vegetation community (Lines 16-20 in the revision).

2. Lines 21-23: Confusing statement. Do you mean this was the average value across five study years? Given the high variability, it may not even be worth stating this mean value, but instead describing the range of variability between the driest year and wettest year.

We have revised this to clarify the difference between wet and dry years, although we have retained average values for the five years (Fig. 4f) to provide an overall context against which to compare the values from individual years (Line 335 in the revision)."

3. Line 24: Results (line 213, Fig. 4F) state that 2018 was a carbon source, not sink.

We have changed the description to "source" in the revision (Line 24 in the revision).

Introduction

4. Line 69: Over what time scale has this happened? The last 10 years, the last 100 years?

We have added a description of the time scale (Lines 70-74 in the revision).

5. Line 72: Can you define the term "semi-fixed"? It is somewhat intuitive but not a common term, bringing to mind less plant cover and more bare soil than I was originally imagining. Perhaps including a range of vegetation basal cover would help.

We have added a definition of semi-fixed sandy land (Lines 78-80 in the revision)."
Zhao, H. L., Zhao, R. L., Zhao, X. Y., Zhang, T. H.: Ground discriminance on positive and negative processes of land desertification in Horqin Sand Land (in Chinese), J. of Desert Research, 28, 8-15. http://210.72.80.159/jweb_zgsm/EN/Y2008/V28/I1/8, 2008.

6. Line 74: Consider replacing "carbon release" and "carbon emission" throughout the text with "carbon source" as the more commonly used term.

We have replaced "carbon release" (everywhere) and "carbon emission" (where appropriate) with "carbon source" throughout the revision.

7. Line 80: Here you allude to natural recovery of sandy land, but this is not referred to again in the last paragraph of the Introduction. It could be an interesting angle for discussion, but it is largely ignored in this manuscript.

We have added a description of natural recovery of the sandy land in the revision (Lines 121-132 in the revision).

Methods

8. Lines 122-124: These methods sentences should be incorporated into the Introduction (last paragraph) to provide a clear statement of all your project aims. How has this restoration affected the ecosystem? What state is it at now – fully recovered, recovering, or still desertified?

We have incorporated these sentences in the Introduction and have added a description of how the restoration has affected the ecosystem (Lines 121-132 in the revision).

9. Lines 125-128: What time range is used to calculate mean annual temperature and precipitation? Given that warming is ongoing, it is important to know the years used to calculate these values.

We have added the time range used to calculate the mean annual temperature and precipitation (i.e., 1960-2014) (Lines 235-236 in the revision).

10. Line 141-144: It is fine to state that more detailed methods are provided in another study, but these references are presumably for other eddy covariance sites, so you should report the eddy covariance equipment and environmental monitoring equipment (instruments, models, manufacturers) plus a brief description of the flux analysis, as a minimum. Your methods should be relatively stand-alone so readers can interpret your results.

We have added Table 1 to provide a detailed description of the instruments we used and their position relative to the soil (Lines 173-175 in the revision). We have also added more details about the eddy covariance analysis at our site (Lines 176-182 in the revision).

11. Line 145: This is less than the range of the average energy balance closure for the global FLUXNET tower network: 0.84 ± 0.2 SE (Stoy et al., 2013, https://www.sciencedirect.com/science/article/pii/S0168192312003413). Why?

The accepted range of the energy balance closure used in the global FLUXNET tower network ranged from 0.56 to 0.97 (Wilson et al. 2002), and the energy balance closure values for our study ranged from 0.58 to 0.67, indicating that the data observed at our study site met the observation requirements (Lines 184-189 in the revision). The Stoy values you cite fall within this range.

Wilson, K., Goldstein, A., Falge, E., Aubinet. M., Baldocchi, D., Berbigier, P., Bernhofer, C., Ceulemans, R., Dolman, H., Field, C., Grelle, A., Ibrom, A., Law, B.E., Kowalski, A., Meyers, T., Moncrieff, J., Monson, R., Oechel, W., Verma, S. Energy balance closure at FLUXNET sites, Agric. For. Meteorol., 113, 223–243, https://doi.org/10.1016/S0168-1923(02)00109-0, 2002.

12. Line 149: What timescale was the Random Forests analysis conducted on? Seasonal mean monthly flux averages? This is important in interpreting results.

We have clarified the timescale (mean daily fluxes) used by the Random Forests model (Line 194 in the revision)

13. Line 154: Your statistical analyses should be fully described here, rather than referring to other papers.

We have added details of the Random Forest analysis (Lines 191-210 in the revision) and of the correlation, ANOVA, and regression analyses (Lines 211-219 in the revision).

14. Line 163: What ANOVA and correlation analysis? Please describe. Also, any testing for normality, transformations, and the threshold p-value should be described.

We have added details of these analyses (Lines 215-217 in the revision) and provide the *P* levels for significance where we report statistically significant results (e.g., the values in tables 2-4).

Results

15. Table 1: The letter is missing from 2021 PPT and is not consistent across the mean row. Logically, the mean row should not have letters, which highlight differences between years here. What is the sampling unit used to describe significant differences among years? Is it monthly means? Daily means? This should be added to the caption and the methods. Also, it would be helpful to add a column for NEE, if not also GPP and Reco, to this table. They are presented in Figure 4F but are useful in Table format. If space is limiting, perhaps the 4 SWC and 4 Ts columns could be limited to 1 column each with the remaining data provided as a Supplementary Table.

Because the *PPT* represents the total value for the whole year, it represents a single data point. Thus, is not realistic to use statistics to analyze the differences between years. We have added the daily means data to describe significant differences among years

and have added *NEE*, *GPP*, and $R_{eco}$ in Table 2 in the revision and have moved the four *SWC* columns and four $T_s$ columns into Supplementary Table S1.

16. Figure 4: Please rescale the x-axis in panels A-E from something like 0.5 to 12.5 so that the bars are more centered.

We have revised the x-axis scale in panels A-E from 0.5 to 12.5 in the revision (Fig. 4 in the revision).

17. Figure 6: In other panels, you have plotted GPP in green and Reco in blue so is there an error in the legend here?

We have unified the legend colors so that we use the same color for a given variable throughout the revision.

18. Figures 7 and 9: Relationships in Panel A appears non-linear, perhaps in some other panels too… have you tested if linear or nonlinear relationship provides a better fit here?

In response to the comments of reviewer 1, we have replaced Figures 7 to 9 with Figure 6 and Table 3, which more clearly show the relationships between the monthly-scale environmental factors and the carbon fluxes during the growing season. Our analysis now focuses on the research hypotheses, and focuses on the effects of drought during the growing season and impacts of the annual-scale values of the environmental factors on carbon fluxes (Lines 245-271, 307-317, Fig. 3, Fig. 6, and Table 3 in the revision).

19. Line 173: Missing decimal in the 2020 PPFD?

We have added the missing decimal (Line 226 in the revision).

20. Line 182: "and" not "but"

We have removed the redundant data and now show only the most important data that is relevant to the study.

21. Line 186-187: It's not necessary to repeat all the information already provided in Table 1, delete. It would be more informative to describe significant differences among years, e.g., wet vs. dry years, or the minimum vs. maximum rainfall.

Because precipitation varied greatly from year to year, listing each year's data more clearly shows the effect of this important variable, so we have retained our detailed description of the precipitation data (Lines 232-238 in the revision).

22. Line 196: "resemble" might be the wrong word here, I'm not sure what this means.

We have replaced "resemble" with "similar" (Line 248 in the revision)

23. Line 206: Replace "not significantly different from the long-term average close to a normal" with "average".

We have changed this to "average" (Line 275 in the revision).

24. Line 239-241: This sentence belongs in Methods, while "The results are summarized in Table 2." is unnecessary and could be added as "(Table 2)" at the end of the relevant sentence(s).

We have revised the description and have analyzed the relationship between monthly environment factors and carbon fluxes of the growing season (Fig 6 and Table 3 in the revision), and conducted deeper analysis focused on the hypothesis, it mainly includes the effects of drought in growing season and long-time scale environmental factors on carbon flux (Lines 245-271, 307-317, Fig. 3, Fig. 6, and Table 3 in the revision).

Discussion

25. Line 261: "ecosystem" might be the wrong word here, I'm not sure what this means.

We have replaced "ecosystem" with "semi-fixed sandy land ecosystem" (Line 330 in the revision). We have also added "recovering" in some places to clarify that this is not a mature, undegraded system.

26. Line 264: "carbon source" and "carbon sink" would be more easily understood terms here.

We have standardized on "carbon source" and "carbon sink" except where "emission" is more appropriate (Lines 331, 334, 339 in the revision).

27. Lines 289-295: Why was PPFD lower in 2020? This is not discussed but should be, as it may (or may not) be helpful in supporting your conclusions here. Is radiation expected to change dramatically in coming years at your site? If not, I question the value of this discussion on the role of solar irradiance in driving daily or seasonal changes in GPP and NEE. Instead, I miss a more detailed discussion on the importance of soil and air temperature, which are important in driving carbon fluxes at your temperate site and are expected to increase with climate change. Based on your dataset, how might this affect your ecosystem?

The reason for the low *PPFD* value in 2022 was an instrument failure, which decreased the measured values. However, this failure occurred during the dormant season (after August). The impact on the carbon flux was therefore small (because the plants were dormant), so for the sake of data integrity, we have retained the data from this period. As you note, solar radiation is not expected to change dramatically, although the predicted future changes in precipitation may lead to changes in cloud cover that affect solar radiation. Speculating about those changes is beyond the scope of this study, as

the changes would not be simple to predict (Lines 227-231 in the revision). We have added a more detailed discussion of the importance of soil and air temperatures for the carbon fluxes (Lines 307-317, 406-430 in the revision) and have described the climate predictions regarding future precipitation and temperatures for our region (Lines 425-430, 465-469 in the revision).

28. Lines 343-345: Awkward sentence, rephrase, could also be more specific. Greater rainfall would allow for higher stomatal conductance, and thus higher photosynthesis and leaf area.

We have revised this and other sentences to be clearer and more specific. For this specific sentence, please see Lines 436-437 in the revision.

Conclusion

29. Lines 357-371: Much of this Conclusion section simply repeats information presented in Results and Discussion, rather than synthesizing the overall importance of your study. Desertification and restoration are not really discussed anywhere but would make an interesting addition to the Discussion/Conclusions. Has the fencing fully restored the ecosystem's function? How many years did (or will) restoration take? Is fencing enough to restore the ecosystem? How much of the landscape is under restoration?

We have revised the Conclusion to focus on whether we confirmed our study hypotheses and how desertification affected the ecosystem. In addition, we have provided more detail about the effects and timescale of restoration in the Introduction (Lines 62-66, 70-74, 86-90, 114-132 in the revision) and the Conclusion (Lines 464-465 in the revision).

30. Lines 373-374: What are the climate predictions regarding precipitation for your region? Is precipitation expected to increase, decrease, or highly uncertain? What about temperature, how will warming affect the ecosystem?

We have added predictions about future precipitation and temperatures for our region and how this will affect the ecosystem in the future (Lines 30-33, 425-430, 467-469 in the revision).

Thank you for your efforts to improve our paper. We hope that our responses and the resulting changes will be acceptable, but we will be happy to work with you to resolve any remaining issues.

Sincerely,

Yuqiang Li, Ph.D.

Northwest Institute of Eco-Environment and Resources

Chinese Academy of Sciences

320 Donggang West Road, Lanzhou, 730000, China

Phone/Fax: 86-931-496-7219

E-mail: liyq@lzb.ac.cn